# The Relationship Between Soccer Participation and Team Cohesion for Adolescents: A Chain-Mediated Effect of Athlete Engagement and Collective Self-Esteem

**DOI:** 10.3390/bs15020155

**Published:** 2025-01-31

**Authors:** Zhihao Zhao, Xiang Che, Haopeng Wang, Yi Zheng, Ning Ma, Liquan Gao, Yizhou Shui

**Affiliations:** 1School of Physical Education, Shaanxi Normal University, Xi’an 710119, China; zhaozhihao@snnu.edu.cn (Z.Z.); 2022303380@snnu.edu.cn (H.W.); zhengyi@snnu.edu.cn (Y.Z.); maning@snnu.edu.cn (N.M.); 2School of Psychology, Shaanxi Normal University, Xi’an 710062, China; xiang.che@snnu.edu.cn

**Keywords:** athlete participation, athlete engagement, collective self-esteem, team cohesion, soccer, adolescents

## Abstract

This study was carried out to explore the mechanism between athlete participation and team cohesion for adolescent school soccer players. We examined the mediating role of athlete engagement and collective self-esteem between athlete participation and team cohesion for adolescent soccer players. A comprehensive investigation of 1659 primary and middle school soccer players was conducted using the Athlete Participation Scale, Athlete Engagement Questionnaire, Collective Self-Esteem Scale, and Team Cohesion Scale, and we conducted correlation, regression and pathway analyses among the variables. We found the following for adolescent soccer players: (1) Athlete engagement plays a complete mediating role between athlete participation and team cohesion. (2) Collective self-esteem plays a complete mediating role between athlete participation and team cohesion. (3) Athlete participation does not directly predict team cohesion, but can influence team cohesion through the chain mediation of athlete engagement and collective self-esteem. This study built a chain mediation model showing that athlete engagement and team cohesion mediated athlete participation and team cohesion, to investigate its mediating role for adolescent soccer players. This study provides theoretical guidance and an empirical basis for the intervention of athlete participation on team cohesion in soccer sports among adolescents.

## 1. Introduction

Soccer is one of the favorite team sports among adolescents. Apart from its role in fostering the physical and emotional well-being of teenagers, it also plays a crucial role in enabling them to manage other aspects of their daily lives, including relationships, employment, and education. Soccer is not just a sport, but it also represents a culture, education, and an atmosphere of social life, and long-term participation in soccer can help individuals quickly integrate into society ([39]).

Cohesion is understood as a “dynamic process that manifests itself in part as a group coming together to pursue a common goal or to satisfy the emotional needs of its members” ([7]). Team cohesion is the ability to draw people to a group and keep them motivated to stay in it; it is also known as “the power to attract members to the team” and “maintaining the motivation in the team” ([33]). Related research has shown that team cohesion research is usually based on preliminary work on the contextual, personal, and leadership correlates of cohesion ([7]). Based on this seminal work, Carron hypothesized that cohesion is a multidimensional structure that distinguishes between individuals and the groups they belong to, as well as task and social dimensions. In a cohesive team, members value their identity within the team and strive to maintain positive relationships with other team members. Relevant researchers have examined team cohesion in sport psychology and other domains, and have found that team cohesion represents the culmination of all the factors that hold a team together ([4]).

Related studies have involved systematic reviews and meta-analyses on adolescent athlete participation, and the results show that the effects of athlete participation on child cognitive function, brain structure and function are beneficial ([37]). Furthermore, it has been shown that anxiety and depression symptoms are significantly lower among adolescents participating in sports than among those who do not participate in sports ([38]). Through soccer participation, teenagers will form peer-leadership structures within the team, which helps to promote the development of the team, improve the communication between the team members, and improve the team cohesion ([41]).

### 1.1. Athlete Participation and Athlete Engagement

Athlete participation is physical and mental engagement in physical education classrooms or extracurricular sports activities, which is usually divided into three dimensions: cognitive participation, emotional participation, and behavioral participation ([1]). Studies suggest that people who participate in sports have more positive body image perception than those who do not ([25]). Athlete engagement is a sustained and relatively stable athletic experience that includes a wide range of positive emotions and the cognitive totality of an individual’s sport, which is characterized by self-confidence, dedication, enthusiasm, and energy in sport ([28]). Overall, athlete participation and athlete engagement are mutually reinforcing and influence each other. By increasing athlete engagement, individuals can improve their sports level and sports achievement, thus enhancing their interest and enthusiasm in sports. Conversely, active participation in physical activities or fitness will also bring higher athlete engagement, making individuals more engaged in exercise.

### 1.2. Athlete Engagement and Collective Self-Esteem

While the association between physical ability and athlete engagement is evident, the relationship between collective self-esteem and athlete engagement is less clear. Some studies have even shown a negative relationship between athlete engagement and self-esteem ([22]). Research has found that “there is little direct evidence that engagement in sports, and winning championships, directly leads to increased self-esteem” ([17]). However, other scholars have demonstrated that individuals who engage in sports have higher self-esteem compared to those who do not ([46]). In soccer, it has been suggested that soccer engagement has been shown to increase athletes’ self-esteem levels, and as self-esteem levels increase, their ability to interact with others is also enhanced ([24]). Thus, there is a positive association between collective self-esteem and athlete engagement, and they reinforce each other. In summary, individuals who are more engaged tend to have higher self-esteem. This is because, through active athlete engagement with soccer, they have increased and built confidence in their abilities and have experienced many challenges and successes with their team members, thus enhancing their collective self-esteem.

### 1.3. Collective Self-Esteem and Team Cohesion

Relevant studies have shown that in team sports, the athletes’ self-esteem improves particularly under the influence of the team ([14]). It has also been demonstrated that among athletes with lower self-esteem, negative emotions are often more prevalent than positive ones ([23]), while athletes with higher self-esteem tend to be able to inhibit negative emotions as well as trivialize negative experiences. Therefore, an increase in players’ self-esteem level may promote team cohesion ([55]). Past research has found that Cohesion, as a major team factor, can better promote collective self-esteem ([54]). If an individual participates in a collective with a higher cohesion, the corresponding individual self-esteem will also be improved. This study concluded that in soccer, players with higher collective self-esteem are more willing to cooperate with each other to achieve the common goals of the team. Secondly, in teams with strong team cohesion, players have more opportunities to learn from their teammates, and can also better understand their own shortcomings, so as to enhance personal confidence and self-esteem. Therefore, hypothesis 1 (H1) is proposed.

**H1:** 
*Athlete engagement and collective self-esteem play a chain mediating role in athlete participation and team cohesion.*


### 1.4. Athlete Engagement and Team Cohesion

Related studies indicate that engagement in sports is a lasting, positive cognitive and emotional experience for athletes ([29]), which contributes positively to improving their athletic performance and ensuring psychological well-being ([18]). Several researchers have argued that team cohesion in soccer is the centripetal force behind soccer players’ trust in each other, satisfying their emotional needs, and achieving the team’s goals under a common goal ([44]). Relevant studies have shown that engagement in soccer has improved the team cohesion and collective efficacy of athletes ([26]). Engaging team sports, like soccer, necessitate a lot of interaction between players during practice and competition. This interaction helps the team develop a sense of accountability and a sense of identity, making each player understand that their contribution is essential to the team’s success, which improves team cohesion. In soccer training and competition, players form common experiences with regard to several aspects, and these common experiences build the foundation of team cohesion ([13]). Consequently, players create strong ties and create teams that are more cohesive by working toward similar goals, overcoming obstacles, and sharing victories and setbacks ([36]). Recent research has suggested that athlete engagement can be utilized as an objective indicator of sports performance and is predictive of team cohesion ([19]). This study, based on previous studies, suggests that players’ athlete engagement and team cohesion are mutually reinforcing relationships in soccer. Only when the players are highly engaged and the team cohesion is strong can a team achieve better results and achieve higher goals. Therefore, H2 is proposed:

**H2:** 
*Athlete engagement plays a mediating role in the influence of athlete participation on team cohesion.*


### 1.5. Athlete Participation and Collective Self-Esteem

Collective self-esteem refers to the feeling and assessment of members of a team regarding the value of a social group, including a racial, ethnic, or working group, etc. ([30]). Prior research has shown that positive mental health, such as an regards an individual’s life satisfaction and well-being, can be a good predictor of their collective self-esteem ([35]). Exercise and Self-Esteem Model (EXSEM) studies have shown that people’s perceptions of body image are closely related to the amount of time spent in athletic activities, and that this mechanism of association is a key factor in influencing their self-esteem ([47]). The skill development hypothesis’s findings indicate a substantial correlation between improvements in self-esteem and adults and adolescents. For example, improvements in perceived physical ability and self-esteem of individuals involved in sports predicted physical self-worth ([6]). Based on the results of previous studies, it is further believed that the active participation of the players helps to create a good team atmosphere, by allowing them to better understand each other and enhance the feelings toward each other. Moreover, players with higher collective self-esteem are more willing to work with teammates because they cherish the unity and harmony of the team. Therefore, H3 is proposed:

**H3:** 
*Collective self-esteem plays a mediating role between athlete participation and team cohesion.*


### 1.6. Athlete Participation and Team Cohesion

Some scholars suggest that people should often work together in groups or teams in order to achieve common goals and then accomplish specific tasks ([3]). As a team sport, soccer is always closely related to teamwork ([9]; [21]; [50]; [60]). One of the most common reasons for participating in soccer is the desire to be part of a team, and the feeling of being part of a team is crucial in this process ([20]). The first lesson in soccer is the importance of teamwork, where “good team players” seems to mean something more noble than “talented players” or even “trained players” ([15]). Providing cohesion as a member of a team is a key determinant of team success and performance ([20]). In soccer, cooperation among teammates is crucial for the team to perform at its highest level, even when some players are autonomous in their work and dependent on others. Therefore, teamwork is a fundamental principle for teams to achieve optimal goals and improve team cohesion ([2]). There are also relevant studies proving that adolescents participating in soccer training or competitions need to unite with their teammates to achieve success, which helps to enhance team cohesion between adolescents ([53]). This study suggests that players’ athletic participation and team cohesion influence each other in soccer. The active participation of the players helps to enhance team cohesion, and the higher team cohesion will stimulate the players’ participation enthusiasm and sense of responsibility, as well as promote team cooperation. Therefore, H4 is proposed:

**H4:** 
*Athlete participation can positively predict team cohesion.*


### 1.7. The Present Study

Although previous researchers have studied the relationship between sports and team cohesion, the relationship between soccer participation and team cohesion has not been fully explored in the previous studies ([2]). Therefore, it is essential to reveal the role of soccer athlete engagement and collective self-esteem from a multi-factor perspective to clarify the relationship between the above characteristics and team cohesion. This study aimed to investigate the relationship between team cohesion and athlete participation in soccer. Specifically, it will investigate the role of athlete engagement and collective self-esteem in this relationship, reveal the impact of soccer on the development of adolescent soccer players, and promote their overall physical health and psychological growth.

## 2. Methods

### 2.1. Participants

A total of 1692 participants from 64 schools across five provinces of China were selected. They all participated in after-school training every day with the school soccer team. After screening, 1659 valid questionnaires were obtained (Table 1), resulting in a 97.0% validity rate. All the participants signed written consent, approved by their parents or guardians prior to filling out the questionnaires. The methods for gathering data respected participants’ rights and followed ethical guidelines. This study was approved by the Ethics Committee of Shaanxi Normal University (No. 202416017).

### 2.2. Procedure

The research data were collected following ethical guidelines to protect the rights of the participants. Formal approval was obtained from the participating schools before administration of the questionnaire. The questionnaires were distributed within the students’ classes under the conditions authorized by each school. Prior to the start of the study, our team members clearly explained the purpose and significance of the study to the participants, as well as the instructions for completing the questionnaire. This was carried out to ensure that the participants understood the voluntary nature of the study and the anonymity of their responses.

#### 2.2.1. Measures

According to our own research goals and needs, we selected the relevant authoritative scale in the field of psychology, and made corresponding adjustments to specific groups and situations on the original basis. We carefully selected the sample to ensure its comprehensiveness, covering towns with different levels of economic development and cities from five different regions. To further reduce the data bias, we also implemented on-site supervision. During the data processing phase, we removed questionnaires with highly consistent answers. And we tested the reliability and validity of all the selected scales.

##### Athlete Participation Scale (APS)

The scale developed by [1] ([1]) was used. The scale was localized and revised by [57] ([57]); it consists of three dimensions: behavioral participation, emotional participation, and cognitive participation. The two dimensions of behavioral participation and emotional participation, each of which is 4 items, and cognitive participation, 5 items, form a scale containing a total of 13 questions. The questions were scored on a 5-point Likert scale, from “completely inconsistent” to “fully consistent” from 1 to 5 points.

##### Athlete Engagement Questionnaire (AEQ)

The AEQ, developed by [28] ([28]), was used to gather data. The AEQ was revised and translated into Chinese by [58] ([58]). The questionnaire comprises 16 questions across four dimensions: self-confidence, vigor, dedication, and enthusiasm. To ensure the relevance of the measurements, the original questionnaire limited the scope of comparison to soccer. The questionnaire used a 5-point Likert scale, ranging from 1 (never) to 5 (always).

##### Collective Self-Esteem Scale (CSS)

The Collective Self-Esteem Scale developed by [59] ([59]) for translating [30] ([30]) was used for the measurement. The scale consists of 16 questions divided into 4 dimensions measured by 4 subscales of collective members’ self-esteem, intrinsic collective self-esteem, extrinsic collective self-esteem, and collective identity, with each subscale containing 4 questions. It was scored using a 5-point Likert scale, with options that transitioned from 1 = completely disagree to 7 = Fully Agree.

##### Group Environment Questionnaire (GEQ)

Team cohesion was measured by the group environment questionnaire (GEQ). The GEQ was first developed by [7] ([7]). The questionnaire then was revised and localized, being translated into Chinese by [31] ([31]). The questionnaire consists of 15 items, which are divided into four dimensions: Individual Attractions to the Group–Task (ATG-T), Individual Attractions to the Group–Social (ATG-S), Group Integration–Task (GI-T), and Group Integration–Social (GI-S). The application guidelines for the questionnaire recommend using the Likert 5-point scale to measure the Chinese adolescent players in practice. Therefore, the original Likert 7-point scale was modified to a Likert 5-point scale, where higher scores indicate better team cohesion.

### 2.3. Data Analysis

The data were analyzed by SPSS 27.0. Firstly, the reliability of the scale was analyzed using Cronbach α. Subsequently, descriptive statistics and correlation analyses were performed. Meanwhile, the mediated model with moderation was tested by Model 6. By generating 95% confidence intervals for the mediating and moderating effects using a sample of 5000 instances, the theoretical hypothesis model was also evaluated. In addition, Amos 28.0 was used to validate the confirmatory factor analysis.

## 3. Results

Since the data in this study were collected using questionnaires, exploratory factor analysis was used to test for possible common method bias. Harman’s one-way test was used, and the results showed that 59 factors were entered, of which 8 had eigenvalues greater than one. The variance explained by the first factor was 25.91%, which was less than the critical value of 40%; therefore, there was no serious common method bias in the data of this study.

This research tested all constructs for validity using confirmatory factor analysis and assessed the reliability of the data using Cronbach’s α. In this research study, the Cronbach’s α coefficient of the APS was 0.971, and the confirmatory factor analysis fit indices were satisfactory: χ^2^/df = 10.783, RMSEA = 0.077, TLI = 0.969, NFI = 0.976, CFI = 0.978, IFI = 0.978. The Cronbach’s α coefficient of the AEQ was 0.948, and the confirmatory factor analysis fit indices were satisfactory: χ^2^/df = 8.422, RMSEA = 0.067, TLI = 0.950, NFI = 0.955, CFI = 0.960, IFI = 0.961. The Cronbach’s α coefficient of the CSS was 0.808, and the confirmatory factor analysis fit indices were satisfactory: χ^2^/df = 1.351, RMSEA = 0.015, TLI = 0.998, NFI = 0.997, CFI = 0.999, IFI = 0.999. The Cronbach’s α coefficient of the GEQ was 0.909, and the confirmatory factor analysis fit indices were satisfactory: χ^2^/df = 4.929, RMSEA = 0.044, TLI = 0.966, NFI = 0.965, CFI = 0.972, IFI = 0.972.

### 3.1. Statistical Description and Correlation Coefficients

Table 2 presents the descriptive statistics of the variables. The correlation analyses revealed significant positive correlations between soccer athlete participation, soccer athlete engagement, collective self-esteem, and team cohesion (all *p*s < 0.01).

### 3.2. Moderated Mediation Model

The results of the mediated effects analysis are shown in Table 3: soccer athlete participation had a direct positive predictive effect on soccer athlete engagement (*β* = 0.205, t = 8.510, *p* < 0.001) and collective self-esteem (*β* = 0.059, t = 2.948, *p* = 0.003); soccer athlete engagement directly and positively predicted collective self-esteem (*β* = 0.597, t = 30.094, *p* < 0.001); when soccer athlete participation, soccer athlete engagement, and collective self-esteem simultaneously predicted team cohesion, soccer athlete engagement and collective self-esteem had significant positive predictive effects (*β* = 0.222, t = 8.197, *p* < 0.001; *β* = 0.329, t = 12.185, *p* < 0.001), while there was no significant predictive effect on team cohesion (*β* = 0.008, t = 0.375, *p* = 0.708); therefore, the mediating variables acted as full mediators, i.e., soccer athlete participation and collective self-esteem played a full mediating role in the effect of soccer athlete engagement on team cohesion.

### 3.3. Chain Mediation Effect Test of the Model

As shown in Table 4 and Figure 1, the mediating effect was tested using the Bootstrap method in the Process macro program, and the results showed that the mediating effect of soccer athlete engagement and collective self-esteem was significant, and the specific analysis showed that the mediating effect was generated through three paths: First, the indirect path of soccer athlete participation → soccer athlete engagement → team cohesion (Bootstrap 95% CI, [0.014,0.035]) suggested soccer athlete engagement significant mediated soccer athlete participation and team cohesion. Second, the indirect path of soccer athlete participation → collective self-esteem → team cohesion (Bootstrap 95% CI, [0.002,0.019]) suggested collective self-esteem significant mediated soccer athlete participation and team cohesion. Third, the indirect path of soccer athlete participation → soccer athlete engagement → collective self-esteem → team cohesion (Bootstrap 95% CI, [0.013,0.030]) suggested soccer athlete engagement and collective self-esteem significantly mediated soccer athlete participation and team cohesion.

## 4. Discussion

This study constructed a chain mediation model to elucidate the association among athlete participation, engagement, collective self-esteem, and team cohesion for adolescent school soccer players in the building of previous studies on the relationship between athlete participation and team cohesion. It was found that athlete participation in soccer can not only have a direct effect on team cohesion, but also have an effect on it through the mediating role of soccer athlete engagement and collective self-esteem. In addition, soccer athlete participation can also positively affect team cohesion through the chain-mediated effects of soccer athlete engagement and collective self-esteem. This is of theoretical and practical significance for improving adolescents’ participation in soccer and their sense of responsibility.

### 4.1. Chain Mediation Role of Soccer Athlete Engagement and Collective Self-Esteem

This study found athlete engagement and collective self-esteem play a chain-mediated role between athlete participation and team cohesion for adolescent school soccer players; as such, H1 is supported. According to this research, teenagers’ team cohesion in soccer may be further enhanced by athlete engagement and collective self-esteem related to player participation.

Research demonstrated the significance of participation and engagement in sports to improve adolescents’ self-esteem ([46]). Opportunities to participate in sports (either now or in the long term) were positively associated with both general and collective self-esteem ([6]). Furthermore, in team sports, where players spend most of their lives training and competing with their teammates and coaches, athletes spend more time living collectively as well as engaging in social interactions with their sports teams compared to other social groups (even families) ([56]). Thus, engaging in team sports is of great significance to collective self-esteem. It is true that people are impacted by their soccer team while they play, and that this interaction fosters a positive sense of self-worth among team members through their positive participation and hard work ([25]). In soccer, the athlete engagement of the players has a significant impact on collective self-esteem. The full participation of the players not only improves their own competitive level, but also enhances the overall strength and team spirit of the team. In the process of common struggle, the players encourage and support each other, forming a strong team cohesion. Every victory is the best proof of the team’s efforts, and also an affirmation of each player’s efforts, which greatly improves the team’s collective self-esteem and confidence. Therefore, it can be found that athlete participation in soccer can positively predict team cohesion through the chain-mediated effects of athlete engagement and collective self-esteem.

### 4.2. Mediating Role of Athlete Engagement in Soccer

This study found that athlete engagement plays a fully mediating role between athlete participation and team cohesion for adolescent school soccer players; thus, H2 is supported. This is consistent with previous findings, as it has been seen that team member performance in soccer has a positive effect on team cohesion ([26]). Additionally, players who perform well on the team or individually show more task cohesion ([51]).

When players are committed to a common goal set by the team, they come closer together, and this shared pursuit of the goal strengthens team cohesion ([10]). In addition, players need to cooperate, support, and work with each other to be successful in a game. In the process of participating in soccer, players will gradually learn how to help each other in times of difficulty and celebrate together in times of success, and this inter-team cooperation and support can help enhance team cohesion ([53]). Nonetheless, similar research discovered that there are notable differences in group cohesion across volleyball teams with lower and higher competitive performance, with more cohesive players often exhibiting superior performance ([43]). In soccer training and competition, the players’ active participation and dedication not only improve the competitive level of the team, but also promote mutual understanding and trust among the players. This experience of working together allows the players to value their connections with each other even more, forming a close team relationship, and thus enhancing team cohesion. Therefore, it can be found that athlete participation in soccer can positively predict team cohesion through the mediating role of athlete engagement.

### 4.3. Mediating Effect of Collective Self-Esteem

This study shows that collective self-esteem plays a fully mediating role in soccer athlete participation and team cohesion, supporting H3. This result suggests that athlete participation can enhance adolescents’ team cohesion by facilitating the development of their collective self-esteem and thus enhancing their team cohesion. This is because collective self-esteem is dependent on the individual’s evaluation of and feelings regarding the importance of the group to which he or she belongs ([30]), closely related to the team, and soccer is the best interpretation of the team. [40] ([40]) was concerned with the phenomenon of collectivism in sports, and believed that a sports team is a cooperative group, and the relationship between people depends on the success or failure of joint activities. This collective action is a projection of societal norms, tasks, and purposes in and of itself.

Previous research has demonstrated that team members with greater collective self-esteem have a higher sense of well-being and responsibility ([35]), which can be a contributing factor in improving team cohesion among adolescents participating in soccer. It was found that psychological collectivism (collective self-esteem) is not only an important manifestation of cohesive groups but also a psychological process that improves team sports performance through cohesion. Strong and steady cohesion not only strengthens collaboration within the team and encourages members to put up maximum effort to accomplish goals, but also contributes to the improvement of members’ feelings of cooperation and winning mindsets ([8]). In addition, social identity theory reflects the importance of group influences and processes in individual behavior. It suggests that individuals define themselves according to their groups and that identifying valued groups enhances self-esteem and self-concept; meanwhile, adherence to group norms is important for individuals to gain group recognition ([45]; [48]). Adolescents who play soccer for their team grow to have a strong sense of team identity, which motivates them to work hard in training and give their all in games. This positive commitment further strengthens the bond between players and can enhance team cohesion among adolescents. Therefore, it can be found that soccer athlete participation in soccer can positively predict team cohesion through the mediating effect of collective self-esteem.

### 4.4. The Effect of Athlete Participation in Soccer on Team Cohesion

The results of this study show that there is no direct relationship between soccer athlete participation and team cohesion, and research hypothesis H4 was not examined. Nevertheless, the conclusions these data reveal are inconsistent with previous research in the same field. Previous research has demonstrated that adolescent athlete participation positively affects team cohesion ([49]). When it comes to teams, younger players are more cohesive than older players ([5]), and shared understanding among team members helps teams achieve better athletic performance, thus enhancing their team cohesion ([13]). Relevant researchers have emphasized the importance of common understanding as the basis of a team’s ability to work together effectively, and shared understanding among team members helps teams achieve better athletic performance, thus enhancing their team cohesion ([52]).

Relevant studies have highlighted the value of team members effectively sharing mental models with one another and observed that this mutual manifestation of team cohesion among players allows one to predict what other members may do in specific situations based on their shared understanding ([16]). In addition, some researchers found that team members must understand how each other performs in specific situations in order to perform effectively together in soccer games ([11]; [12]). According to the present study, team cohesion is crucial for the growth of shared knowledge and enhanced team performance in team sports like soccer, where players rely on each other for success ([32]).

Although the results of this study cannot confirm the direct impact of athlete participation on team cohesion, according to the orientation of the data, we believe that athlete participation has a positive effect on team cohesion through the mediation of athlete engagement and collective self-esteem. Secondly, soccer being used, as one of the most typical team sports, which also reflects the diversity of participants, may affect the deviation of the data. However, relevant studies have proved that team cohesion is associated with coaching behavior ([27]), this may be one of the reasons for the non-significant direct effects. Furthermore, it have also demonstrated that the leadership style and behavior of coaches have a great influence on team cohesion ([42]). Therefore, on the basis of previous studies, this study further explores that athlete participation not only has a direct impact on team cohesion, but also has positive effects through the two variables of athlete engagement and collective self-esteem. Playing soccer has a complex impact on team cohesion since the sport not only improves player collaboration and trust off the field but also builds team spirit and cohesion during practices and team-building exercises. Team duties, individual motivation, and the degree to which individuals strive for team success are among the variables that influence how much soccer is played and how cohesive a team is.

## 5. Limitations and Future Directions

It should be noted that there are two limitations in this study. First, this study aimed to comprehensively explore the influence mechanism of physical activity enjoyment achieved from soccer on team cohesion by utilizing a large sample size that encompasses various educational stages. To more accurately investigate the influence mechanism in future research, a sample from a specific educational stage could be adopted. Second, This study adopts cross-sectional data, which may have a certain impact on causal judgments ([34]). In future research, longitudinal data can be used to further explore the relationships between variables.

## 6. Conclusions

This study built a chain mediation model to investigate the mediating role between adolescent athlete participation in soccer and team cohesion. This study suggested that athlete engagement plays a fully mediating role between athlete participation in soccer and team cohesion. Collective self-esteem plays a fully mediating role between soccer athlete participation and team cohesion. Soccer athlete participation does not directly predict team cohesion, but can influence team cohesion through the chain mediating role of soccer athlete engagement and collective self-esteem. In soccer, coaches should stimulate their enthusiasm, focus on personal growth, provide personalized guidance, create a positive atmosphere, encourage cooperation, and set common goals to enhance team cohesion.

## Figures and Tables

**Figure 1 behavsci-15-00155-f001:**
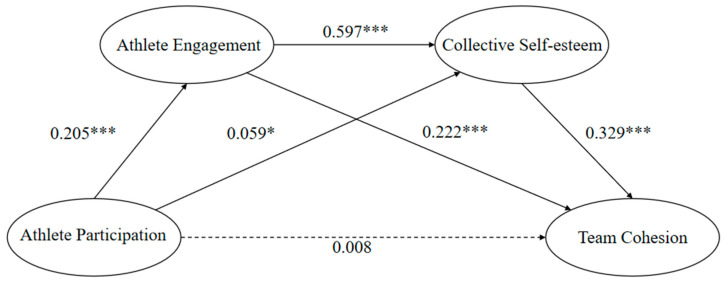
Path analysis of the mediating effects of athlete engagement and collective self-esteem on team cohesion. *: *p* < 0.05; ***: *p* < 0.001.

**Table 1 behavsci-15-00155-t001:** Demographic information.

Variables	Numbers	Percentages	Variables	M ± SD
Gender				
Male	1135	68.415%	Weekly training hours (in hours)	
Female	524	31.585%	Primary	6.986 ± 4.851
Phase of study			Junior high	7.007 ± 4.846
Primary	885	53.345%	Senior high	7.467 ± 4.949
Junior high	543	32.731%	Training years (by years)	
Senior high	231	13.924%	Primary	3.853 ± 2.387
Place of abode			Junior high	3.870 ± 2.391
City	1398	84.268%	Senior high	4.158 ± 2.513
Village	261	15.732%		

**Table 2 behavsci-15-00155-t002:** Correlation analysis among all variables.

Variables	*M*	*SD*	1	2	3	4
1. APS	4.092	0.961	1			
2. AEQ	4.196	0.656	0.205 ***	1		
3.CSS	5.551	0.751	0.181 ***	0.609 ***	1	
4. GEQ	4.511	0.457	0.113 ***	0.424 ***	0.465 ***	1

NOTE: APS = athlete participation scale; AEQ = athlete engagement questionnaire; CSS = Collective Self-esteem Scale; GEQ = Group Environment Questionnaire. The Mean (M) and Standard Deviation (SD) are calculated by dividing the total score by the individual item of scale. ***: *p* < 0.001; same below.

**Table 3 behavsci-15-00155-t003:** Regression analysis between variables.

Outcome	Predictive Variables	R^2^	F	β	t
AEQ	APS	0.041	72.446	0.205	8.510 ***
CSS	APS	0.374	496.099	0.059	2.948 *
	AEQ			0.597	30.094 ***
GEQ	APS	0.247	181.812	0.008	0.375
	AEQ			0.222	8.197 ***
	CSS			0.329	12.185 ***

Note: AEQ = athlete engagement questionnaire; CSS = Collective Self-esteem Scale; GEQ = Group Environment Questionnaire. *: *p* < 0.05; ***: *p* < 0.001.

**Table 4 behavsci-15-00155-t004:** Pathway analysis.

Path	Effect Value	SE	Bootstrap 95% CI		Relative Mediating Effect
Ind1	0.023	0.005	0.014	0.035	43.39%
Ind2	0.010	0.004	0.002	0.019	18.44%
Ind3	0.021	0.004	0.013	0.030	38.36%

Note: Boot standard error, Boot CI lower limit and Boot CI upper limit refer to the standard error, lower limits and upper limits of the percentile Bootstrap method, respectively.

## Data Availability

The data that support the findings of this study are available from https://doi.org/10.6084/m9.figshare.25858510 (accessed on 27 December 2024).

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
