# Peer review of "The Relationship Between Soccer Participation and Team Cohesion for Adolescents: A Chain-Mediated Effect of Athlete Engagement and Collective Self-Esteem"

_behavsci, 2025, doi:10.3390/bs15020155_

Round 1
Reviewer 1 Report
Comments and Suggestions for Authors
I hope this letter finds you well. I had the opportunity to review your article titled, “The Relationship between soccer participation and team cohesion for Adolescents: A Chain-Mediated Effect of athlete engagement and collective self-esteem”, which was submitted to Behavioral Sciences.
Title: Please capitalize each word in the title.
1. Abstract
The research methods were missing from the abstract. Briefly describe your research methods.
The sentences use past and present tenses. I recommend that you match the verbs in each sentence.
2. Introduction
The researcher's argument is not revealed in the introduction. Only the contents of the previous research were written in report format. It is necessary to logically explain the researcher's claims based on prior research.
Line 92. Not suitable for academic format. Long sentences tend to make the meaning ambiguous. Write in short sentences so that the author's argument is conveyed accurately.
Previous studies have very well organized hypotheses on the relationships between the study variables (player participation and player promise, player promise and self-esteem, self-esteem and team cohesion, etc.).
3. Methods
Table1 This does not conform to the academic format. The research procedures, research tools, and data analysis were deemed appropriate to analyze the purpose of the study. It is recommended that the validity and reliability of the research tool be presented in the research methods. The description of research ethics is missing. Additional description is required. The table format throughout the paper has been edited differently from the academic conference format. The tables need to be edited.
4. Results
The results well explain what was required by the research method.
Tables and figures are used to help readers better understand the results of the study.
5. Discussion
The discussion in this study is considered to be well organized with respect to the hypothesis set by the researcher. The purpose of the study was well analyzed, and the researcher's argument was logically presented based on similar previous studies. However, the researcher's argument is judged to be somewhat insufficient compared to the analysis and presentation of prior research.
6. Limitations and future research
I also think that the suggestions for follow-up research are excellent, and I think this part can be moved to the conclusion.
7. Conclusion
The conclusion is a summary of the research and the areas in which the results of this study can provide academic and empirical support. From this perspective, the researcher adequately explains what the results of this study can provide academically or empirically. Acknowledgments and Funding are overlapping. Please write only one.
8. Reference
I would like you to provide the DOI for the prior research. The entire bibliography must be revised to conform to the academic journal format.
Author Response
Comments 1: Please capitalize each word in the title. |
Response 1: Thank you for pointing this out. We agree with this comment. Therefore, we have Modify the title to “The Relationship Between Soccer Participation And Team Cohesion For Adolescents: A Chain-Mediated Effect Of Athlete Engagement And Collective Self-esteem”.
|
Comments 2: The research methods were missing from the abstract. Briefly describe your research methods. |
Response 2: Agree. We have described the study method in the abstract “ A comprehensive investigation of 1,659 primary and middle school soccer players was conducted using the Athlete Participation Scale, Athlete Engagement Questionnaire, Collective Self-Esteem Scale, and Team Cohesion Scale, and conducted correlation, regression and pathway analyses among the variables.” on line 16.
Comments 3: The sentences use past and present tenses. I recommend that you match the verbs in each sentence. Response 3: Thank you for pointing this out. We agree with this comment. Therefore, we matched the verbs in each sentence.
Comments 4: The researcher's argument is not revealed in the introduction. Only the contents of the previous research were written in report format. It is necessary to logically explain the researcher's claims based on prior research. Response 4: Thank you for pointing this out. We agree with this comment. we elaborate the arguments in this study based on previous studies and add the following: “ This study concluded that in soccer, players with higher collective self-esteem are more willing to cooperate with each other to achieve the common goals of the team. Secondly, in teams with strong team cohesion, players have more opportunities to learn from their teammates, and can also better understand their own shortcomings, so as to enhance personal confidence and self-esteem. ” on line 106. “ This study based on previous studies suggested that players' athlete engagement and team cohesion are mutually reinforcing relationships in football. Only when the players are highly engaged and the team cohesion is strong, can the team achieve better results and achieve higher goals. ” on line 131. “ Based on the results of previous studies, it is further believed that the active participation of the players helps to create a good team atmosphere, so as to better understand each other and enhance the feelings between each other. Moreover, players with higher collective self-esteem are more willing to work with teammates because they cherish the unity and harmony of the team. ” on line 151. “ This study suggests that players' athlete participation and team cohesion influence each other in football. The active participation of the players helps to enhance the team cohesion, and the higher team cohesion will stimulate the players' participation enthusiasm and sense of responsibility, and promote the team cooperation. ” on line 175.
Comments 5: Line 92. Not suitable for academic format. Long sentences tend to make the meaning ambiguous. Write in short sentences so that the author's argument is conveyed accurately. Response 5: Agree. we changed the 92 lines into short sentences “ Therefore, an increase in players' self-esteem level may promote team cohesion ”.
Comments 6: Previous studies have very well organized hypotheses on the relationships between the study variables (player participation and player promise, player promise and self-esteem, self-esteem and team cohesion, etc.). Response 6: Agree. The study clarified the relationship between the variables in the introduction section and proposed four hypotheses for this study.
Comments 7: Table1 This does not conform to the academic format. The research procedures, research tools, and data analysis were deemed appropriate to analyze the purpose of the study. It is recommended that the validity and reliability of the research tool be presented in the research methods. The description of research ethics is missing. Additional description is required. The table format throughout the paper has been edited differently from the academic conference format. The tables need to be edited. Response 7: Thank you for pointing this out. We agree with this comment. Therefore, we have explained the reasons for selecting all the variables in the study tool section, and also explained the good reliability and validity of the variables in the results section. Furthermore, the description of research ethics is given on line 197: “All the participants signed written consents approved by their parents or guardians prior to filling out the questionnaires. The methods for gathering data respected participants' rights and followed ethical guidelines. This study was approved by the Ethics Committee of Shaanxi Normal University (NO. 202416017).” We have also standardized the format of all of the tables in the article.
Comments 8: The results well explain what was required by the research method. Tables and figures are used to help readers better understand the results of the study. Response 8: Thank you for pointing this out. We agree with this comment. According to the previous research paradigm, the paper first analyzed the correlation between the variables, then established the regression equation model for each variable, conducted the regression analysis for each variable, and finally analyzed each path in the model. All the data are presented in tabular and graphic ways to understand the relationship between variables.
Comments 9: The discussion in this study is considered to be well organized with respect to the hypothesis set by the researcher. The purpose of the study was well analyzed, and the researcher's argument was logically presented based on similar previous studies. However, the researcher's argument is judged to be somewhat insufficient compared to the analysis and presentation of prior research. Response 9: Agree. According to your suggestion,Compared with previous studies, we have added some own arguments to the article: “ In soccer, the athlete engage of the players has a significant impact on the collective self-esteem. The full participation of the players have not only improved their own competitive level, but also enhanced the overall strength and team spirit of the team. In the process of common struggle, the players encourage and support each other, forming a strong team cohesion. Every victory is the best proof of the team's efforts, and also the affirmation of each player's efforts, which greatly improves the team's collective self-esteem and confidence. ” on line 356. “ In the training and competition of soccer, the players' active participation and dedication not only improve the competitive level of the team, but also promote the mutual understanding and trust among the players. This experience of working together allows the players to value the connection with each other even more, forming a close team relationship, thus enhancing team cohesion.” on line 380.
Comments 10: I also think that the suggestions for follow-up research are excellent, and I think this part can be moved to the conclusion. Response 10: Thanks for your valuable advice, We decided to summarize and refine the recommendations for subsequent studies and moved them to the conclusion.
Comments 11: The conclusion is a summary of the research and the areas in which the results of this study can provide academic and empirical support. From this perspective, the researcher adequately explains what the results of this study can provide academically or empirically. Acknowledgments and Funding are overlapping. Please write only one. Response 11: Thank you for pointing this out. We agree with this comment. Acknowledgments and Funding are overlapping, we removed the Acknowledgments.We also added some feasibility opinions to our conclusions:“ In soccer, coaches should stimulate their enthusiasm, focus on personal growth, provide personalized guidance, create a positive atmosphere, encourage cooperation, and set common goals to enhance team cohesion.” on line 474.
Comments 12: I would like you to provide the DOI for the prior research. The entire bibliography must be revised to conform to the academic journal format. Response 12: According to your suggestion,We checked the DOI of previous studies and revised the entire bibliography to conform to the academic journal format. |

Reviewer 2 Report
Comments and Suggestions for Authors
1. Introduction
• Issue: The introduction lacks detailed references to recent studies on team cohesion and athlete engagement.
• Suggestion: Include recent research, such as meta-analyses or systematic reviews, on adolescent sports participation and its psychological effects.
• Proposed Addition: “Recent studies (e.g., [cite 2020–2024 literature]) highlight the critical role of engagement and self-esteem in fostering cohesion within youth sports teams.”
2. Methods
• Issue: The justification for using self-reported measures is not sufficiently detailed.
• Suggestion: Discuss the limitations of self-reporting and explain how potential biases were mitigated (e.g., using validated scales).
• Proposed Addition: “Although self-reported measures may introduce bias, the use of validated scales like the APS and AEQ ensures reliability and construct validity.”
3. Results
• Issue: Figure and table captions are not descriptive enough to be interpreted independently.
• Suggestion: Add more detail to captions, explicitly stating key findings.
• Proposed Addition: “Figure 1. Path analysis of the mediating effects of athlete engagement and collective self-esteem on team cohesion. Significant paths are indicated (p < 0.05).”
4. Discussion
• Issue: The discussion does not sufficiently address the non-significant direct effect of athlete participation on team cohesion.
• Suggestion: Provide a deeper analysis of why this relationship was not observed and compare it to prior studies.
• Proposed Addition: “The non-significant direct relationship may stem from contextual factors, such as differences in coaching styles or team dynamics, as suggested by [cite relevant studies].”
5. Conclusion
• Issue: The conclusion reiterates findings but lacks emphasis on actionable strategies for educators and coaches.
• Suggestion: Include specific recommendations for integrating engagement-building activities into training programs.
• Proposed Addition: “Coaches should incorporate team-building exercises that enhance collective self-esteem, such as group goal-setting and shared challenges, to foster team cohesion.”
6. Formatting and Language
• Address minor grammatical inconsistencies, such as verb tense shifts in the introduction.
• Simplify overly complex sentences for better readability.
Author Response
Comments 1: Issue: The introduction lacks detailed references to recent studies on team cohesion and athlete engagement. |
Response 1: Thank you for pointing this out. We agree with this comment. Therefore, we have added a portion of the most recent related studies.
|
Comments 2: Suggestion: Include recent research, such as meta-analyses or systematic reviews, on adolescent sports participation and its psychological effects. |
Response 2: Agree. According to your suggestion, We add to recent research on sports participation in meta-analyses or systematic reviews adolescents and its psychological impact: “ Related studies have conducted systematic review and meta-analysis on adolescent athlete participation, and the results show that the effects of athlete participation on child cognitive function, brain structure and function are beneficial(Owen et al., 2022). Furthermore, it has been shown that anxiety and depression symptoms are significantly lower among adolescents participating in sports than among those who do not participate in sports(Panza et al., 2020). Through soccer participation, teenagers will form peer-leadership in the team, which helps to promote the development of the team, improve the communication between the team members, and improve the team cohesion(Piasecki, 2024).”on line 54.
Comments 3: Proposed Addition: “Recent studies (e.g., [cite 2020–2024 literature]) highlight the critical role of engagement and self-esteem in fostering cohesion within youth sports teams.” Response 3: Thank you for pointing this out. We agree with this comment. Therefore, We added to recent literature to illustrate the key role of athlete engagement in improving team cohesion: “Recent research has suggested that athlete engagement can be utilized as an objective indicator of sports performance and is predictive of team cohesion(Gu & Xue, 2022)” on line 131.
Comments 4: Issue: The justification for using self-reported measures is not sufficiently detailed. Suggestion: Discuss the limitations of self-reporting and explain how potential biases were mitigated (e.g., using validated scales). Proposed Addition: “Although self-reported measures may introduce bias, the use of validated scales like the APS and AEQ ensures reliability and construct validity.” Response 4: Thank you for pointing this out. We agree with this comment. Therefore, we added an interpretation about self-reported in the Methods section:“According to our own research goals and needs, we selected the relevant authoritative scale in the field of psychology, and made corresponding adjustments to specific groups and situations on the original basis. We carefully selected the sample to ensure its comprehensiveness, covering towns with different levels of economic development and cities from five different regions. To further reduce the data bias, we also implemented on-site supervision. During the data processing phase, we removed those questionnaires with highly consistent answers. And we have tested the reliability and validity of all the selected scales.” on line 213.
Comments 5: Issue: Figure and table captions are not descriptive enough to be interpreted independently. Suggestion: Add more detail to captions, explicitly stating key findings. Proposed Addition: “Figure 1. Path analysis of the mediating effects of athlete engagement and collective self-esteem on team cohesion. Significant paths are indicated (p < 0.05).” Response 5: Agree. According to your suggestions, we have modified the description of all figures and tables, and will standardize the format.
Comments 6: Issue: The discussion does not sufficiently address the non-significant direct effect of athlete participation on team cohesion. Suggestion: Provide a deeper analysis of why this relationship was not observed and compare it to prior studies. Proposed Addition: “The non-significant direct relationship may stem from contextual factors, such as differences in coaching styles or team dynamics, as suggested by [cite relevant studies].” Response 6: Thank you for pointing this out. We agree with this comment. Therefore, new related studies we added to the discussion illustrate potential causes of non-significant direct effects: “ However, relevant studies have proved that team cohesion is associated with coaching behavior(Light Shields et al., 1997), this may be one of the reasons for the non-significant direct effects. Furthermore, it have also demonstrated that the leadership style and behavior of coaches have a great influence on team cohesion(Ramzaninezhad & Keshtan, 2009). ” on line 446. ”
Comments 7: Issue: The conclusion reiterates findings but lacks emphasis on actionable strategies for educators and coaches. Suggestion: Include specific recommendations for integrating engagement-building activities into training programs. Proposed Addition: “Coaches should incorporate team-building exercises that enhance collective self-esteem, such as group goal-setting and shared challenges, to foster team cohesion.” Response 7: Thank you for pointing this out. We agree with this comment. Therefore, we added some feasibility suggestions to the coach in the conclusion section: “ In soccer, coaches should stimulate their enthusiasm, focus on personal growth, provide personalized guidance, create a positive atmosphere, encourage cooperation, and set common goals to enhance team cohesion.” on line 474.
Comments 8: Address minor grammatical inconsistencies, such as verb tense shifts in the introduction. Simplify overly complex sentences for better readability. Response 8: Thank you for pointing this out. We agree with this comment. Therefore, we have addressed all the grammatical problems in the article to improve sentence readability. |
